# Lithium-Ion Batteries under Low-Temperature Environment: Challenges and Prospects

**DOI:** 10.3390/ma15228166

**Published:** 2022-11-17

**Authors:** Hanwu Luo, Yuandong Wang, Yi-Hu Feng, Xin-Yu Fan, Xiaogang Han, Peng-Fei Wang

**Affiliations:** 1State Grid East Inner Mongolia Electric Power Supply Co., Ltd., Hohhot 010010, China; 2Center of Nanomaterials for Renewable Energy, State Key Laboratory of Electrical Insulation and Power Equipment, School of Electrical Engineering, Xi’an Jiaotong University, Xi’an 710049, China

**Keywords:** lithium-ion battery, low temperature, electrode materials, electrolyte, kinetics

## Abstract

Lithium-ion batteries (LIBs) are at the forefront of energy storage and highly demanded in consumer electronics due to their high energy density, long battery life, and great flexibility. However, LIBs usually suffer from obvious capacity reduction, security problems, and a sharp decline in cycle life under low temperatures, especially below 0 °C, which can be mainly ascribed to the decrease in Li^+^ diffusion coefficient in both electrodes and electrolyte, poor transfer kinetics on the interphase, high Li^+^ desolvation barrier in the electrolyte, and severe Li plating and dendrite. Targeting such issues, approaches to improve the kinetics and stability of cathodes are also dissected, followed by the evaluation of the application prospects and modifications between various anodes and the strategies of electrolyte design including cosolvent, blended Li salts, high-concentration electrolyte, and additive introduction. Such designs elucidate the successful exploration of low-temperature LIBs with high energy density and long lifespan. This review prospects the future paths of research for LIBs under cold environments, aiming to provide insightful guidance for the reasonable design of LIBs under low temperature, accelerating their widespread application and commercialization.

## 1. Introduction

Since their commercialization in 1991, LIBs have been widely applied in portable electronic devices, stationary energy storage, space exploration, subsea operation, and military defense owing to their high energy density, long life span, high operating voltage, light weight, and great environmental compatibility, even monopolizing some energy storage fields [1,2,3,4,5,6,7,8,9,10,11]. With the development of science and technology, great efforts to reduce the carbon consumption have led to a change in renewable energy, and the use of electric systems is growing rapidly. Thus, higher demands have been put forward for LIBs under extreme environments such as high temperatures above 40 °C and cold conditions below 0 °C. At present, the commercial LIBs based on an ethylene carbonate (EC) electrolyte and graphite anode still encounter poor performance at low temperature, with deterioration and failure becoming major obstacles. When the temperature drops below 0 °C or lower, limited by the reduced conductivity and the solidification of electrolyte, the capacity degrades rapidly, whereby commercial LIBs can only maintain a small portion of their capacity or even stop working. At low temperature, the polarization becomes larger, and the discharge voltage decreases accordingly, resulting in severe energy loss which cannot meet the requirement in application. Simultaneously, the Li^+^ (de)intercalation process is restricted in cold conditions, leading to lower coulombic efficiency and the difficulty in charging and discharging, further deteriorating the life span of LIBs. Moreover, the serious Li dendrites that grow on the surface of the anode during low-temperature charging can even cause safety issues such as thermal runaway. These dilemmas severely limit the practicality of LIBs in low temperatures [8,12,13,14,15,16,17,18,19]. Fortunately, external secondary heating strategies and thermal management can effectively raise the local temperature to keep the LIBs operating, and such methods have been employed in many LIB devices, especially large-scale energy storge systems. Nevertheless, approaches relying on external devices inevitably result in additional energy consumption and higher cost [12,20,21,22,23,24,25]. Therefore, searching for satisfactory LIBs in terms of battery chemistry, particularly those with high energy density and fast charging capability, is urgent. It is also of great significance for the widespread application of LIBs and the field of electrochemical energy storage and conversion. Recently, low-temperature LIBs are of intense interest and have attracted abounding research; various modification methods for electrode, new anode materials, and novel design ideas of electrolytes make it possible solve the problems under low temperature. In light of such advances, it is necessary to perform a critical review for this promising field.

As “rocking-chair batteries”, LIBs work via the (de)intercalation of Li^+^ between the cathode and anode, involving diffusion in both solid and liquid phases. During the charging process, Li ions deintercalate from the cathode and move to the anode through the electrolyte. When discharging, Li^+^ ions undergo an opposite process and eventually return to the cathode [26,27]. Although LIBs have been widely commercialized as an important energy storage device, further enhancement of energy density and safety demands are still the key problems encountered, especially in extreme temperature environments. When working at high temperature (above 40 °C), more side reactions and elemental dissolutions will occur due to higher reactivity between the electrolyte and electrodes. At the same time, the reduction in electrolyte becomes severe and thickens the solid electrolyte interphase (SEI), thus enlarging the polarization and deteriorating the electrochemical properties. However, some solvents with a low flash point may cause troubling fires or even explosions, resulting in serious safety concerns [28]. However, the challenges for LIBs at low temperature are more severe and urgent. On the basis of the above mechanism of LIBs, intrinsic diffusion restrictions inevitably affect the electrochemical performance, while the kinetics is the fundamental reason at subzero temperatures. Therefore, as the temperature is lowered, the diffusion of Li^+^ inside the cells is definitely limited. This reduced Li^+^ transport rate will induce severe polarization, inhibit the phase transition on the electrode, and worsen the reaction process, as well as the capacity and rate performance [29,30,31,32,33,34]. Meanwhile, the influence of low temperature on LIBs is also exhibited on the electrode–electrolyte interfaces. With the decrease in temperature, SEI will become thicker due to excessive overpotential. However, when Li^+^ moves through the SEI film, it needs to overcome a higher energy barrier, which limits the transport of Li^+^ and leads to the increase in charge transfer resistance (R_ct_). In addition, the viscosity of the electrolyte increases or even causes solidification under low temperature, which deteriorates the compatibility of the electrolyte with the separator and electrode, further affecting the formation of the cathode electrolyte interphase (CEI) and SEI. For instance, the EC-based commercial electrolyte cannot be utilized below 0 °C. Worst of all, low-temperature environments may exacerbate the Li plating on the anode side, as well as promote the growth of Li dendrites and “dead” Li during cycling [28,35,36]. Such deficiencies would lead to local volume expansion, destroy the internal structure of LIBs, and cause internal short-circuits and thermal runaway, seriously threatening the safety of LIBs (Figure 1).

The possible reasons for the undesirable performance of LIBs at low temperatures can be briefly summarized as follows: (i) the poor kinetics on both the interphase and the electrodes, which means larger SEI resistance and a reduction in the Li^+^ diffusion coefficient in the cathode and anode; (ii) decreased ionic and electronic conductivity, lower viscosity, and high freezing point of the electrolyte; (iii) Li plating and Li dendrites on the surface of the anode, which threaten the safety and cycle life of LIBs. In designing a battery which can adapt to cold environments, numerous studies have been performed recently on battery chemistry. Various electrode modification techniques have been delivered to optimize the conductivity and Li^+^ transport properties, such as the surface coating, nanosizing, morphological designing, and element doping. Simultaneously, several anode materials have been developed and implemented in LIBs, taking advantage of their respective strengths to inhibit the deterioration from different aspects. Furthermore, strategies related to the electrolyte such as low-freezing-point cosolvents, high-concentration electrolytes, blended salts, and additives have been proposed to compensate for the deficiency at low temperature.

Despite the fact that the electrochemical low-temperature performance has been improved as a result of above approaches, the deterioration at low temperature is still a serious hurdle for LIBs. To facilitate future endeavors in addressing the dilemmas outlined above, we organized this review to encourage more effective design strategies in low-temperature LIBs. In this review, we summarize the relevant scientific problems and mechanisms of low-temperature LIBs, conclude the recent research progress and achievements from the aspects of cathode, anode, and electrolyte, and then examine the optimization methods that may overcome the barriers at low temperature. We deliver our prospects and suggestions for the improvement methods at low temperature, with the aim of determining the key toward realizing energy storage in extreme conditions and providing reliable guidance in terms of research directions for the development of low-temperature LIBs.

## 2. Challenges Facing Cathode Materials under Low Temperature

LIBs are influenced by various components during operation; as the core part of LIBs, the cathode is the center of Li^+^ exchange. The properties of LIBs are strongly dictated by the Li^+^ transport properties, structure stability, electrochemical reversibility, and Li^+^ storage capacity of the cathode, especially at low temperature [37,38]. Degradation of the cathode at low temperature is mainly due to the decreased Li^+^ diffusion coefficient and high R_ct_ caused by low kinetics, leading to significantly increased polarization. These problems impede the (de)lithiation process, incurring certain energy and capacity loss [8,19,39]. Therefore, in order to optimize the performance at low temperature, especially to improve the ionic/electron conductivity and structure stability in the cathode side, various modifications on LIBs cathodes, including surface coating, particle modification, and element doping have been extensively studied.

### 2.1. Surface Coating

A small amount of surface coating on the cathode can obviously improve the conductivity at low temperature, reduce the cell impedance, and inhibit side reactions between the electrode and electrolyte, thereby effectively enhancing the charge/discharge capacity and energy density of LIBs at low temperature. Considered to be the most effective coating materials, carbon-based materials have been widely investigated and applied to various cathodes; a simple schematic diagram is shown in Figure 2a [40,41,42,43,44]. Since they have been proven to improve the low-temperature performance, a variety of carbon compounds have been combined with different cathodes. Chen and coworkers [45] used KB carbon as the source and coated it onto Li_3_V_2_(PO_4_)_3_ (LVP) to synthesize a nanocomposite. The strategy enhanced the Li^+^ transport properties and protected the structure of the material, resulting in a discharge capacity up to 92 mA·h·g^−1^ under −30 °C with a 4 C rate. At −20 °C and a 0.1 C rate, the LiFePO_4_/C delivered by Li’s group also exhibited a high discharge capacity up to 109.2 mA·h·g^−1^ [46]. Recently, several novel coating strategies have been proposed to further exert the advantages of carbon coating. For instance, structures with carbon coating have been designed in different ways. Ning et al. [47] used a carbon nanotube (CNT) suspension to produce the cathode slurry, and then applied LiFePO_4_ in a well-dispersed CNT conductive network. The results showed that a modest concentration of 0.2% CNT could effectively suppress the enhancement in overpotential and improve the conductivity at low temperature. The half-cell delivered a specific capacity of about 90 mA·h·g^−1^ at −10 °C and a current rate of 1 C. Moreover, Xie’s group [48] successfully developed LiFePO_4_/C with a bridge network utilizing graphene nanofibers (Figure 2b). At −20 °C, the composites with 5 wt.% graphene nanofibers (G−5) provided a high discharge capacity of 124.4 mA·h·g^−1^ at 0.1 C and showed 92% capacity retention at 1 C after cycling 200 times (Figure 2e). Wu and coworkers [49] successfully synthesized LiFePO_4_/C with carbon sources of fructose and calcium lignosulfonate via a hydrothermal process. When operated at −20 °C, the capacity retention of this cathode was 20% higher than that of samples synthesized using the solid-state method.

Oxide coatings are also propitious to the low-temperature electrochemical properties of LIB cathodes. For example, Li_2_O–B_2_O_3_ coated on LiNi_1/3_Co_1/3_Mn_1/3_O_2_ (NCM) significantly reduced the R_ct_. Such a design enhanced the discharge capacity at −20 °C from 37.2 to 101.9 mA·h·g^−1^, keeping a high retention of 93.4% after operating for 50 cycles [18]. Although oxide coatings may provide certain benefits to the cathode, most oxides need to be used in conjunction with a carbon coating to facilitate the contact between the coating and cathode or to improve the stability. Wang and coworkers [50] coated 3 wt.% MnO_2_ onto an LVP/C composite (M−3); the MnO_2_ and C exhibited a complementary distribution on the surface and formed a continuous conductive network, which reduced the charge transfer hurdle on the surface of electrode, thereby improving the conductivity of electrons and Li^+^ (Figure 2c,d). As illustrated in Figure 2f, when operating under −20 °C, the capacity of M−3 increased significantly at 0.2 C and 1 C, accompanied by lower polarization compared to the pristine LVP/C (M−0). CeO_2_ has demonstrated efficiency in enhancing low-temperature performance. CeO_2_-modified LVP/C and LiFePO_4_/C both confirmed that CeO_2_ possesses the ability to inhibit the increase in R_ct_ and heighten the diffusion coefficient of Li^+^, which can be employed in the field of rapid charging at low temperature [51,52]. In addition, according to the study performed by Yang et al. [53], 3 wt.% Li_4_Ti_5_O_12_ (LTO) coating on LiFePO_4_/C material could effectively reduce the R_ct_ from 1862.0 to 622.1 Ω. In addition to carbon and oxides, additional substances have been explored as coatings. Zhao et al. [54] synthesized Li_1.2_Ni_0.13_Co_0.13_Mn_0.54_O_2_ samples coated with 2% AlF_3_; the capacity at −20 °C of the coated cathode was approximately doubled compared to the original sample. Conductive polymers with a similar mechanism to carbon but more compatible with the electrolyte were also investigated to modify the cathode. LiFePO_4_ coated with 2.95% polypyrrole showed an initial capacity of 128 mA·h·g^−1^ at 0.1 C and −20 °C, being about 83.7% of the capacity tested at 25 °C [55].

### 2.2. Element Doping

External element doping is seen as a facile but effective strategy to improve the electrochemical performance of the cathode by exerting an ideal synergistic effect in the material [32,56,57,58]. Wei et al. [39] designed the Li_1.16_[NiMnCo]_0.327_Mo_0.02_O_2_ material by doping the Li and Mo elements. The introduction of Li and Mo in bulk formed a Li-rich structure, delivering an enhancement of electronic conductivity and structure stability. The optimized sample performed satisfactorily at −30 °C, maintaining a capacity of 97 mA·h·g^−1^ and 65.4 mA·h·g^−1^ at 0.3 C and even 5 C, respectively. Li’s group [59] replaced the Fe element with Ti, preventing the growth and agglomeration of particles, forming more uniform spherical particles, and successfully heightening the capacity from 87.3 to 112.8 mA·h·g^−1^ at −20 °C. By doping the B element into carbon-modified Li_1.2_Mn_0.54_Ni_0.13_Co_0.13_O_2_, Li and coworkers enlarged the interplanar spacing via stronger TM–O bonds, optimizing the kinetics at the low temperature and inhibiting the migration of transition-metal atoms during charging (Figure 3a). When cycled at −20 °C, the cathode exhibited 108.9 mA·h·g^−1^ (0.1 C), and 91.9% of the capacity could be maintained after 100 cycles, much higher than that of the pristine cathode (75.61%) (Figure 3b) [60]. Cui and coworkers [61] incorporated P into LiFePO_4_/C, providing higher specific capacity and coulombic efficiency. This P-doped cathode showed 82.7 mA·h·g^−1^ discharge capacity and initial coulombic efficiency up to 73.3% at −40 °C and 0.1 C, whereas the unmodified LiFePO_4_ only delivered 68.7 mA·h·g^−1^ and 61.5% in the same experimental conditions. Kou’s group [62] found that the doping with trace amounts of Co would reduce the activation energy of the charge transfer process, thereby enhancing the electrochemical performance at ultralow temperature. After 1000 cycles, The Cr-doped LiNi_0.45_Cr_0.1_Mn_1.45_O_4_ proposed by Wang’s group delivered high capacity over 100 mA·h·g^−1^ below 0 °C [63]. In addition, surface doping of Al has been proven to be beneficial to the NCM cathode under low temperature [64].

### 2.3. Other Modification Methods

The physical properties of cathode particles are related to their performance at low temperature. It has been found that a reduction in particle size not only provides short Li^+^ diffusion pathways and the kinetic properties of the material, but also enlarges the specific surface area, leading to better contact between the electrode and electrolyte, thereby further improving the transport property of lithium ions. Thus, small particles can effectively overcome the limitations caused by a low diffusion coefficient under low temperature [65,66]. LiFePO_4_ with different particle sizes exhibited distinct electrochemical properties at low temperature [67]. When tested at −20 °C and 0.2 C, the capacity of the LiFePO_4_ with D50 of ~1.01 μm reached 112.6 mA·h·g^−1^, while the samples with 1.52 μm only provided 78.2 mA·h·g^−1^. By reducing the particle size of the material, the discharge capacity increased by about 44%. Furthermore, the purity of electrode samples also has an effect on the low-temperature performance. Zhao’s work [67] also compared the performance of industry-grade LiFePO_4_ (In-LFP) with low purity and reagent-grade LiFePO_4_ (Re-LFP) with high purity. The results suggested that the low-purity LiFePO_4_/C cathode performed at only 40–60% capacity at −20 °C compared to the Re-LFP. Although In-LFP had a lower R_ct_ due to the presence of highly conductive impurities, these impurities could hinder Li^+^ diffusion or even distort the lattice, resulting in a low Li^+^ diffusion coefficient, which was two orders of magnitude smaller than that of the pure one (Figure 3c,d). Qin et al. [68] synthesized a pure-phase LVP with high-rate capability. When tested at −10 °C and 10 C after 5000 cycles, 80% of the capacity could be retained due to high Li^+^ conductivity and optimum structural stability. Furthermore, the designed morphology of the cathode material manifests effective improvement of the structural stability and the diffusion kinetics. In addition to directly influencing the performance such as by enlarging the specific surface area, modification of the morphology can optimize the low-temperature properties. For instance, the coral-shaped LiFePO_4_/graphene prepared by Fan’s group exhibited a morphology with many holes inside (Figure 3e,f); such holes acted as chambers for the Li^+^ transport and intercalation, as well as helped the carbon to be better coated on the cathode, thereby leading to great low-temperature performance. When working in a cold environment of −40 °C, the half-cell could deliver a low capacity decay of 0.066% per cycle and a great Coulombic efficiency of about 99% (Figure 3g) [69].

As mentioned above, numerous efforts have recently been made in the field of low-temperature LIBs cathode, realizing certain achievements. Nevertheless, diverse cathodes still encounter several problems such as a low diffusion coefficient, as well as side reactions between the cathode and electrolyte. The majority of studies focused on strategies aimed at enhancing the transport property, as well as the conductivity and the stability of the cathode. External surface coating and internal element doping are currently considered promising directions. However, there are certain challenges when using such methods. Surface coatings show the attractive ability to enhance the low-temperature electrochemical properties of cathodes when using a small amount. However, the introduction of coating materials may result in the formation of byproducts, such as Fe_2_P in the LiFePO_4_ system, which would lead to unavoidable capacity sacrifice and contamination of purity [70]. Considering the limitations above, more attention should be paid to the synergy of multiple approaches in the future. The notable low-temperature properties of different cathodes are listed in Table 1. Additionally, whether at room temperature or at low temperature, a high energy density is consistently the goal. Enlarging the voltage window is regarded as a useful approach; however, such research in the field of low temperature still needs further studies.

## 3. Challenges Facing Anode Materials under Low Temperature

Numerous studies have confirmed that the performance of low-temperature LIBs depends more on the (de)intercalation kinetics on the anode surface. At low temperature, anodes are restricted with low electronic and ionic conductivities, poor Li^+^ diffusion ability, increased R_ct_, and limited desolvation kinetics; however, more serious problems are the Li plating and Li dendrites occurring at the anode [71,72,73]. A suitable design of the anode can significantly overcome the deterioration of LIBs at low temperature. Boosting the Li^+^ diffusion coefficient or Li^+^ desolvation ability is an effective method to optimize the electrochemical performance.

### 3.1. Carbon-Based Anodes

#### 3.1.1. Graphite

At present, the current anode materials are carbon-based, among which graphite is the most common. When working at low temperature, the Li^+^ diffusion capability deteriorates, and the conductivity of SEI is reduced correspondingly, resulting in poor Li^+^ (de)intercalation ability. Thus, a portion of Li^+^ is deposited on the surface of the material instead of being properly inserted into the structure of graphite. Notably, the potential range of Li^+^ intercalation in graphite is only about 0.05–0.2 V, close to the reduction potential of Li^+^ (0 V). Moreover, the overcharging process induced by low temperature would decrease the operating voltage of graphite, severely aggravating the deposition of Li^+^ (Figure 4a). The deposited lithium shows strong activity and directly reacts with the electrolyte, accompanied by dramatic volume expansion, leading to a loss of capacity. However, the Li deposited on the surface of SEI would form projections; as the reaction continues and the inhomogeneity increases, a portion of the projections can grow into dendrites, which may pierce the separator and cause a short-circuit inside the cell (Figure 4b,c). These issues dramatically impact the performance and safety of LIBs at low temperature. In addition, the bottom part of the dendrites usually reacts with the electrolyte first, causing the front part to fall off and become “dead lithium”. Graphite’s weak rate capacity also limits its application at low temperature, especially at high current density [73,74]. Moreover, unacceptable thermal effects also represent an obstacle at low temperature. Lithiation on graphite is an endothermic process; thus, when the temperature drops, the decrease in external heat inhibits the desolvation of Li^+^ during charging, thereby hindering the lithiation of graphite material. To cope with the challenges at low temperature, several methods have been applied to carbon-based anodes [35]. Similar to the approaches related to the cathode, coating and element doping are considered promising strategies to optimize the diffusion of Li^+^. Recently, Lu and coworkers [75] added N element into graphite to form a highly branched nitrogen-doped graphitic (BNG) tubular; the N was distributed evenly in the framework, leading to a larger interslab distance and better conductivity. When tested at −20 °C, the capacity of BNG reached 135.8 mA·h·g^−1^, exhibiting promising low-temperature performance. The oxidized graphite with Cu surface coating developed by Mancini et al. reached a high capacity of 103 mA·h·g^−1^ at 0.2 C and under −30 °C, driven by the catalytic effect of Cu, which effectively reduced the polarization and R_ct_ [76]. Similarly, Nobili’s group [77] coated a thin Sn layer on the oxidized graphite; this anode showed lower R_ct_ at low temperature, and it could deliver a high capacity of up to 152 mA·h·g^−1^ at −30 °C and 0.2 C, whereas the pristine samples lost almost all capacity.

#### 3.1.2. Graphene

In addition to graphite, various carbon-based materials with different microstructure have been developed as promising anodes for low-temperature LIBs. Graphene has gained great attention for its nanostructure, which can effectively optimize the performance. Raccichini et al. [80] proposed a type of multilayer crystalline graphene using a special multistep approach; this structure could reduce the size and the thickness of particles. During low-temperature cycling under −30 °C and 0.05 A·g^−1^, the improved carbon anode provided a higher capacity of 130 mA·h·g^−1^ compared with 25 mA·h·g^−1^ for commercial graphite. Furthermore, the approach of doping was also tested to modify the carbon anode. Li’s group [6] prepared a N-doped TiO_2_/TiN/graphene nanocomposite using precursors of oxidized TiN and graphene, which achieved 211 and 73 mA·h·g^−1^ capacity under conditions of 0.1 and 5 A·g^−1^ at −20 °C. Such successful outcomes exhibit the exciting application prospects and further optimization potential of graphene.

#### 3.1.3. Carbon Nanotubes

Extensive research efforts have been invested in CNT for its high conductivity, large specific surface area, and great stability compared to other carbon-based materials. Xu’s group [79] synthesized a novel porous graphite nanosheet (PGN)/CNT composite, where PGN played a role in shortening the Li^+^ diffusion paths and increasing the Li^+^ intercalation sites through the additional pores in the graphite, while CNT effectively prevented the restacking of sheets (Figure 4d–f). Under such improvements, when cycled at −40 °C and 0.1 C, the electrode retained satisfactory capacity of 180 mA·h·g^−1^, whereas commercial graphite anode almost ceased to operate under the same conditions. Hu and coworkers [81] proposed an LTO/Ag/CNT anode which exhibited great low-temperature electrochemical properties. Following the replacement of the common binder with CNT, the electrode became smoother and offered more sites for Li^+^ insertion (Figure 5a), suppressing the negative effect of the common binder. The electrode provided 140 mA·h·g^−1^ capacity at the ultralow temperature of −60 °C and a rate of 0.2 C (Figure 5b).

#### 3.1.4. Other Carbon-Based Anodes

Carbon nanofiber (CNF) is also considered as an alternative anode at low temperature for LIBs. Li and coworkers [83] introduced the Fe element into CNF to form a homogeneous hybrid Fe/Fe_3_C/CNFs anode using the electrospinning method. The optimized samples showed much better electrochemical performance than the bare CNFs due to the good cabalistic effect. After operating at −15 °C and 200 mA·g^−1^ for 50 cycles, the reversible capacities of Fe/Fe_3_C/CNFs and CNFs were 380 and 80 mA·h·g^−1^, respectively. In addition to anode modification, the prelithiation strategy has been considered. Liu’s group [84] delivered a novel prelithiation method for a hard carbon (HC) anode, which was coupled with LVP and initially cycled to form the prelithiated Li_x_C anode. The prelithiated material can be operated even at an ultralow temperature of −40 °C. However, due to the complex steps of prelithiation, it is still not suitable for commercialization, necessitating further research in the future. Moreover, other carbonaceous materials such as soft carbon and porous carbon have also been developed and are seen as potential candidates. However, they have not exhibited satisfied competitiveness at low temperature compared with graphite, necessitating further investigation.

### 3.2. Ti-Based Anodes

Although graphite has been proven to offer considerable advantages in LIBs, it is difficult to implement on a broad scale due to serious issues at low temperature. Hence, non-carbon materials have gained a great deal of attention. Among all, TiO_2_ anodes have been extensively studied due to their low cost, great stability, and high operating voltage (1.5–1.7 V vs. Li/Li^+^), which can effectively avoid Li deposition at low temperature. Meanwhile, the small volume expansion of TiO_2_ also helps to eliminate the capacity decay caused by strain, which is very helpful in the synthesis of many other materials [85,86,87]. Additionally, Ti-based spinel LTO also exhibits great low-temperature performance with good reversibility of Li (de)intercalation and a small volume effect [72]. Although it also encounters serious capacity loss at low temperature, the unique advantages of LTO lead to great potential for application in cold conditions. The operating voltage of 1.55 V vs. Li/Li^+^ effectively prevents the formation of Li dendrites, albeit at the cost of capacity compared to graphite, supporting exceptional safety at low temperature. Ti-based anodes suffer from low conductivity and a low Li^+^ diffusion coefficient; however, several studies have improved the electrochemical performance, which is expected to be further optimized. At present, nanoscale titanium-based anodes have been explored. Park’s group [88] investigated the effect of LTO size on performance at −30 °C, revealing that LTO with smaller particles had superior ionic transport properties. Pohjalainen et al. [89] discovered a similar tendency upon synthesizing an LTO with a secondary particle size of 0.5–5 μm and a specific surface area of 7 m^2^·g^−1^ through intensive grinding, which could increase the capacity at 1 C and −20 °C by 30%. Unfortunately, Ti-based materials are limited by their conductivity at low temperature, whereas carbon materials possess certain advantage of conductivity. Therefore, carbon coating and element doping have also gained attention to improve the properties of Ti-based anodes. Li et al. [90] designed LTO/biomass-derived carbon microspheres, introducing a 3 wt.% conductive carbon on the structure. This method optimized the contact between the electrolyte and anode with a unique porous structure, enhancing the conductivity of the anode. The LTO/biomass-derived carbon exhibited a capacity of 173 and 150 mA·h·g^−1^ at −10 and −20 °C with a rate of 1 C. The peapod-like LTO with a carbon shell reported by Peng and coworkers demonstrated a large surface area, improved conductivity, and a shorter Li^+^ diffusion path, showing better rate capability and higher capacity. The half-cell could provide a satisfactory discharge capacity of about 122 mA·h·g^−1^ even at 30 C under −25 °C, whereas the micro-LTO only maintained a capacity of about 24 mA·h·g^−1^ [91].

### 3.3. Alloying Anodes

#### 3.3.1. Silicon

Silicon as a representative alloy anode material is suitable for application at low temperature. When employed as an anode, Si shows a different mechanism to intercalation-type anodes. During the charging process, Li^+^ and Si are alloyed into Li_x_Si instead of intercalated, effectively alleviating the severe problem of poor intercalation kinetics at low temperature. Furthermore, Si anodes with an outstanding theoretical capacity over 4000 mA·h·g^−1^ show an incomparable advantage at low temperature [35,92,93,94,95,96]. Such a high capacity can be ascribed to the large Li^+^ accommodation (3.75 Li^+^ per atom) compared with graphite (1/6 Li^+^ per atom). Markevich’s group [82] examined the electrochemical performance of Si and graphite under the same conditions (Figure 5c,d). The results demonstrated that the replacement of graphite with Si could dramatically heighten the capacity, and such optimization became more obvious at low temperature. Subburaj’s group [97] assembled a silicon/LiNi_0.8_Co_0.15_Al_0.05_O_2_ pouch cell to further explore its cryogenic properties. When tested at −40 °C, a high discharge capacity over 700 mA·h·g^−1^ was delivered, equivalent to 65% of that at 20 °C. Nevertheless, the Si anode endures a large volume effect (even up to nearly 300%) during operation. This undesirable volume change can lead to serious pulverization and SEI breakage, repeatedly exposing the anode surface to the electrolyte and resulting in cell failure [98]. To realize the full application potential of the Si anode, several methods, such as nanostructuring and fabricating porous structures, have been developed to suppress the serious volume changes, which are expected to be implemented at low temperature.

#### 3.3.2. Other Alloying Anodes

As a member of the same family of alloy anodes, Sn and Ge anode also possess outstanding theoretical capacity of 994 and 1624 mA·h·g^−1^, respectively. Unfortunately, they are also restricted by the capacity degradation induced by mechanical stress during cycling [94,99,100]. Although Sn and Ge have similar properties to Si, they are more used as components in composite anodes. A Ge anode was optimized by Choi et al. via constructing the mesoporous structure, and it demonstrated high capacity over 1400 mA·h·g^−1^ at −20 °C. When coupled with LiFePO_4_, the LiFePO_4_/Ge full battery exhibited great retention up to 80% at 0.5 C and −20 °C after 50 cycles [101]. Such results suggest that the porosity can effectively improve the performance of LIBs at low temperature, which is worthy of further exploration. Recently, alloying anodes have received widespread attention, but they are far from achieving their high theoretical capacity in practical applications. Even if several advances have been realized to reduce the volume effect of alloying materials, they are still in the exploration stage and waiting for further optimization [35,102,103,104].

Regarding low-temperature anodes, Li platting and Li dendrite cannot be ignored as a result of the low operating voltage of graphite-based carbon materials. Despite methods such as external coating, nanostructure, element doping, and prelithiation being able to address the limitations to a certain extent, the low-temperature performance is still inadequate to meet the requirements. The electrochemical properties of the anodes mentioned in above are outlined in Table 2. In addition to the commonly employed carbon anodes, several promising materials with higher capacity and voltage which favor the properties under cold environment have been developed. Ti-based anodes which can eliminate the risk of Li dendrites and the instability of SEI show immense promise due to their practicality at low temperature. Unfortunately, their low energy density and high operating voltage lead to severe restrictions, necessitating further research. Furthermore, alloying anodes manifest promising potential in the field of low-temperature anodes because of their extremely high theoretical capacity and special alloying process, whereas they encounter serious volume effects, and related studies remain limited. Overall, non-carbon anode materials are not mature enough for deployment, but they exhibit wide prospects in a low-temperature environment.

## 4. Challenges Facing Electrolyte under Low Temperature

As a prominent component of LIBs, the electrolyte plays a crucial role from the macroscopic to the microscopic dimensions in terms of conductivity, solvation structure, and film-forming ability, which are significant for the Li^+^ transport properties between electrodes. At temperatures below 0 °C, the viscosity of the electrolyte increases while the Li^+^ conductivity decreases, limiting the process of Li^+^ diffusion. Noteworthily, the low-temperature restriction degree of Li^+^ transport in the electrolyte is much larger than that in the electrode, inducing more serious challenges. In addition to the effect on transport, the deficiency at low temperature is also exhibited at the electrode–electrolyte interface. The wettability and compatibility of the viscous electrolyte with respect to the electrode and separator become worse, increasing the resistance of LIBs, while a thicker SEI on the anode side hinders transport on the interface, as well as the desolvation process. Such deterioration may even induce the formation of Li dendrites. Furthermore, the choice of electrode is dictated by the electrochemical stability window of the electrolyte, which is one of the decisive factors of energy density. A desired low-temperature electrolyte not only needs to meet the requirements of high Li^+^ conductivity, a broad electrochemical window, stable chemical properties, and good safety, but should also possess weak Li^+^ affinity to facilitate Li^+^ desolvation, ensuring diffusion in the SEI and electrode [105,106]. EC has been widely used as a fundamental solvent component of LIB electrolyte owing to its great film-forming ability and great Li^+^ transportation capability. Unfortunately, the high freezing point (36.4 °C) and intrinsically high desolvation barrier of EC dramatically restrict its application at low temperature [107]. Hence, there is an increasing need and desire for reliable solvents in low-temperature environments. Presently, several studies have been performed to find a solvent with high dielectric constant and low viscosity. A high dielectric constant plays a role in providing a great ability to dissociate the salt, promoting the Li^+^ solvation process and further enhancing the transfer kinetics of Li^+^, whereas a low viscosity can suppress the decrease in the Li^+^ transport coefficient at low temperature and reduce the interfacial resistance. Unfortunately, these two characteristics are difficult to achieve in a sole substance, because an enhancement of dielectric constant results in higher viscosity, which is not conducive to the diffusion of Li^+^. Striking a balance between the dielectric constant and the viscosity is, therefore, vital in designing the ideal electrolyte.

### 4.1. Solvents

#### 4.1.1. Cosolvent Electrolytes

To minimize the limitations of EC, various cosolvents with a low freezing point and low viscosity have been added to replace the EC to form a combined system which is conductive to the low-temperature electrochemical performance. Due to their low freezing point and low viscosity, linear carboxylates such as methyl acetate (MA), methyl butyrate (MB), methyl formate (MF), ethyl acetate (EA), ethyl propionate (EP), and propyl butyrate (PB) are considered cosolvents for low-temperature electrolyte. Such solvent molecules with polar functional groups are beneficial to the dissociation of lithium salts, thereby improving the conductivity of Li^+^. Notably, when used with a graphite anode, carboxylate solvents can optimize the properties of the SEI, achieving good reversible ability at low temperature. Logan et al. [108] explored an MA-based electrolyte and compared it with a common carbonate-based electrolyte; when assembled in a LiNi_0.5_Mn_0.3_Co_0.2_O_2_/graphite full battery and tested at −10 °C and 1 C, the MA-based electrolyte showed 62.5% retention of its room-temperature capacity, higher than the 13% retention of the conventional binary solvent. Xu’s group blended MA as a cosolvent (50 vol.%) with carbonates and obtained a novel electrolyte with high conductivity which adapted to the cold environment of −60 °C. Similarly, Zhang and coworkers [109] added MB to design a 1 M LiNF_6_S_2_C_2_O_4_ (LiTFSI) diethyl carbonate (DEC)/EC/MB (1:1:1) electrolyte, which dramatically diminished the viscosity from 52.07 mPa·s to 13.75 mPa·s while increasing Li^+^ conductivity to 3.44 mS·cm^−1^ from 1.38 mS·cm^−1^ at −20 °C. As demonstrated in Figure 6a,b, the SEI film became compact with the introduction of MB and led to more uniform Li plating. However, linear carboxylates encounter undesirable decomposition when the temperature rises, incurring a serious impact on the rate performance and life span, indicating that they are not suitable for a wide temperature range.

Despite a higher freezing point than linear carboxylates, carbonate-based cosolvents have still gained much attention because of their good stability. Janak et al. [110] investigated the performance of 20 different combinations of solvent compositions; the results demonstrated that reductions in cyclicity and chain length were conducive to the low-temperature performance, whereby the combination of EC/propylene carbonate (PC)/ethyl methyl carbonate (EMC)/dimethyl carbonate (DMC) (1.8:0.3:3:3.5) exhibited the greatest capacity retention at −20 °C and −30 °C. A solvent can be rationally designed using different compositions to achieve low temperature objective. Zhang’s [111] group investigated the performance of several cosolvents at low temperature and proposed a novel type of decimal solvent-based electrolyte by utilizing a high entropy effect (Figure 6c). The decimal solvent possessed a satisfactory low freezing point below −130 °C (Figure 6d–f). Such an electrolyte provided a high Li^+^ conductivity of 0.62 mS·cm^−1^ at −60 °C, showing a desired room-temperature capacity retention of 80% at −40 °C when employed with the LiMn_2_O_4_/LTO electrode. In addition to the above studies which strived to reduce the amount of EC by introducing a cosolvent, some studies proposed an EC-free electrolyte to meet the low-temperature requirements. Petibon et al. [112] proposed a 1 M LiPF_6_ methyl propionate (MP)/vinylene carbonate (VC) (95:5) electrolyte; the NCM/graphite pouch cell assembled with this electrolyte operated well at −14 °C and a current rate of 4 C.

#### 4.1.2. Fluorinated Solvents

At low temperature, the desolvation process in LIBs is considered as one of the main kinetic barriers, and it is significantly correlated with the solvation structure. As the temperature decreases, the desolvation energy gradually increases, strengthening the affinitybetween Li^+^ and solvent; such a strong interaction inevitably inhibits the desolvation process. Electronegative F atoms are added into the polar solvent molecules to suppress the problems. Methyl 3,3,3-trifluoropionate (MTFP), ethyl trifluoroacetate (ETFA), 2,2,2-trifluoroethyl butyrate (TFEB), methyl 2,2,2-trifluoroethyl carbonate (FEMC), fluoromethane (FM), methyl trifluoroacetate (MTFA), and trifluoroethyl n-butyrate (TFENB) have been explored to soften the attraction with Li^+^ and diminish the energy obstacles, thus improving the performance at low temperature [113,114]. Holoubek’s group [115] reported an all-fluorinated carboxylate ester-based electrolyte of 1 M LiPF_6_ MTFP/FEC (9:1); when assembled with the LiNi_0.8_Co_0.1_Mn_0.1_O_2_ (NCM811) half-cell, it showed great capacity of about 161, 149, and 133 mA·h·g^−1^ under −40, −50, and −60 °C, respectively, far exceeding that of the common commercial electrolyte. Fan and coworkers [116] introduced a fluorinated carbonate to highly fluorinated nonpolar solvents and prepared the 1.28 M LiNF_2_S_2_O_4_ (LiFSI) FEC/FEMC/tetrafluoro-1-(2,2,2-trifluoroethoxy)ethane (D2) electrolyte; when cycled with this electrolyte at −20 °C and 1/3 C rate, the LiNi_0.8_Co_0.15_Al_0.05_O_2_ half-cell performed at a high capacity of up to 150 mA·h·g^−1^ for 450 cycles, while the cell with a commercial electrolyte could only provide 35 mA·h·g^−1^ for 100 cycles under the same conditions. Notably, the batteries could deliver 96 mA·h·g^−1^ even at an ultralow temperature of −85 °C.

#### 4.1.3. Other Novel Solvents

In addition to common carbonates and carboxylates, several novel solvent components have been investigated to replace traditional commercial solvents. Hu’s group [117] proposed isoxazole (IZ) as a new electrolyte solvent at low temperature due to its high dipole moment, freezing point of −67 °C, and low viscosity. The Li^+^ conductivity of 1 M LiPF_6_ EC/IZ (1:10) could reach 15 mS·cm^−1^ at −20 °C, threefold that of the common EC/EMC-based electrolyte. When mixed with FEC, the Li/graphite cell using 1 M lithium difluoro(oxalato)borate (LiDFOB) IZ/FEC (10:1) delivered a reversible capacity of 187.5 mA·h·g^−1^ and 120 mA·h·g^−1^ at −20 °C and −30 °C (0.1 C), respectively, while the capacity of the cell using the EC/EMC-based electrolyte was only 23.1 mA·h·g^−1^ and 10 mA·h·g^−1^, respectively. It was shown that the 0.9 M LiDFOB sulfolane (SL)/dimethyl sulfite (DMS) (1:1) electrolyte had stable cycling performance, great Li^+^ transport properties, and satisfactory film-forming capability. The introduction of DMS strengthened the conductivity and stability of SEI due to its weak affinity with Li^+^, bringing about lower desolvation energy. When employed in the LiFePO_4_ half-cell, a discharge capacity of around 80 mA·h·g^−1^ was maintained at −20 °C and 0.5 C [118]. Xu et al. [72] developed a 1,3-dioxane (DIOX)-based electrolyte; the 0.75 M LiTFSI DIOX electrolyte exhibited a freezing point lower than −100 °C. Even at an ultralow temperature of −80 °C, the LTO half-cell employing the 0.75 M LiTFSI DIOX electrolyte retained 60% of its room-temperature capacity with a 0.1 C rate. Moreover, Thenuwara et al. [74] proposed an electrolyte with 1,3-dioxolane (DOL)/1,2-dimethoxyethane (DME) as the solvent, which exhibited good ionic conductivity (0.4 mS·cm^−1^) at −80 °C. Furthermore, nitriles such as acetonitrile, butyronitrile, and propionitrile have gained interest over recent years for their ability to form thin and uniform interphases, and they have been studied as cosolvents. However, nitrile-based electrolytes suffer from toxicity, poor safety, and low compatibility with anodes, hindering their wide application [106,119,120,121].

### 4.2. Lithium Salts

#### 4.2.1. Alternative Li Salts for LiPF_6_

As a vital component of the electrolyte, lithium salt is regarded as a key to tuning the LIB properties at low temperature. The conductivity and SEI are strongly influenced by the solubility and dissociation degree of Li salt [122,123]. The desired Li salt should have the ability to be dissolved and decomposed in the solvent, while the anions on the electrode need to be stable to avoid side reactions. LiPF_6_ is the most used Li salt in commercial electrolytes, which is widely employed due to its comprehensive properties. Nevertheless, the corrosiveness of LiF and HF decomposed by LiPF_6_ and its poor performance at low temperature restrict its wide implementation. Compared with LiPF_6_, LiBF_4_ salt with a smaller anionic radius and stronger binding capacity can deliver a lower R_ct_, and it has been applied at low temperature. Unfortunately, as one of the alternatives to LiPF_6_, LiBF_4_ encounters issues of corrosion and an unsatisfactory film-forming ability, unable to meet the requirements for LIBs at low temperature. LiBC_4_O_8_ (LiBOB) has also been investigated due to its great film-forming capability, but its high viscosity and low solubility in carbonate seriously deteriorate its practicality at low temperature when used in LIBs [124]. Considering the advantages of LiBF_4_ and LiBOB, a number of studies mixed the two salts to make a dual-salt electrolyte. Although such electrolytes can be used at low temperature, their capacity was not satisfactory. Furthermore, LiDFOB has been proposed due to its special structure, being composed of semi-molecules of both salts. LiDFOB combines the advantages of LiBF_4_ and LiBOB, being able to form a HF-free SEI layer on the anode during cycling, thus effectively inhibiting the formation of Li dendrites and further improving the electrochemical properties under low temperature. For example, the study performed by Tan et al. applied LiDFBO as the Li salt, achieving a great outcome at −70 °C [117]. Nonetheless, LiODFB still suffers from similar solubility difficulties to LiBOB, and it is expected to be intensively studied. LiTFSI and LiFSI have gained great attention due to their desired thermal stability and low R_ct_ [72,116,125]. For example, LiTFSI and LiFSI were employed as the salt in diethyl ether (DEE) and ETFA solvents; the electrolytes were tested at low temperature and performed well. However, the TFSI^−^ and FSI^−^ anions may corrode the Al collector, seriously deteriorating the performance of LIBs at both room temperature and low temperature [28,34,106,126]. In light of the above analysis, although several Li salts are being researched in the process of finding alternative salts, the dominance of LiPF_6_ still cannot be shaken. Blending several types of Li salts to achieve a synergistic effect is considered a promising approach to optimize the performance of LIBs in cold environments [105,126]. Lv and coworkers [16] blended LiBF_4_ and LiPF_6_ to produce a series of blended-salt electrolytes, and the dual-salt electrolyte of 0.3 M LiBF_4_ + 0.7 M LiPF_6_ DMC/EMC/butyl acetate (BA)/EC delivered the best properties at low temperature; when assembled with a NCM811 half-cell, it maintained 64% room-temperature capacity even at −40 °C, demonstrating a significant improvement compared to any single-salt electrolyte (Figure 7a).

#### 4.2.2. High-Concentration Electrolyte and Localized High-Concentration Electrolyte

During the operation of LIBs, Li^+^ ions are transported with a solvation sheath structure as demonstrated in Figure 7b. As the concentration of salts increases, more solvent molecules participate in the composition of the solvation sheath, while the number of free anions is increased, thus leading to an SEI which contains more inorganic components such as LiF and Li_2_O. The structure with rich inorganic components can enhance the density of SEI and effectively passivate the anode, facilitating the uniform and quick transportation of lithium ions. Moreover, the high concentration of salt may also result in a significant decrease in the freezing point of electrolyte, which is promising for low-temperature applications. Nevertheless, it is obvious that the inevitable high viscosity of high-concentration electrolytes (HCEs) further limits their performance when the temperature decreases. To overcome such a shortcoming, an effective strategy is to introduce an inert solvent to dilute the high-concentration electrolyte to form a localized high-concentration electrolyte (LHCEs) (Figure 7c). Notably, the inert diluent can reduce the viscosity and promote the transport properties of Li^+^ without participating in the structure of the solvation sheath or affecting the solvation structure, as shown in [105,128,129,130,131,132]. Dichloromethane (DCM) was used as the diluent in 5 M LiTFSI EA/DCM (1:4) electrolyte due to its low viscosity (0.44 MPa s) and low freezing point (−95 °C); the optimized electrolyte efficiently solved the low-viscosity problem and showed a great conductivity of 0.6 mS·cm^−1^ even at −70 °C [132]. Ren et al. [129] designed a high-concentration electrolyte of LiTFSI/TMS/1,1,2,2-tetrafluoroethyl-2,2,3,3tetrafluoropropyl ether (TTE) (1:3:3), where TTE was used as the diluent; when using this electrolyte at −10 °C and 0.2 C, the NCM/Li cell delivered a specific capacity of over 100 mA·h·g^−1^. Feng’s [133] group prepared a localized high-concentration electrolyte by adding the fluorination diluent 1,1,2,2-tetrafluoroethyl methyl ether into the MA-based solvent. When employed in an LNMO/Li battery at 0.2 C and an ultralow temperature of −50 °C, the cell retained 80.85% of its room-temperature capacity, exhibiting promising prospects in high-voltage and low-temperature applications. Likewise, Holoubek et al. [134] developed a localized high-concentration electrolyte of LiFSI/DME/bis(2,2,2 trifluoro ethyl)ether (BTFE) with different concentrations; the NCM811 half-cell could retain 63% of its room-temperature capacity at −60 °C using this electrolyte, and it possessed excellent capacity retention without capacity loss for over 100 cycles. In the field of Li salts, studies have mainly focused on anion species due to their considerable influence on the degree of dissociation and their relationship with the SEI composition. On the other hand, the number of studies using the salt approach is comparatively smaller due to the limited choices of anions at low temperature.

### 4.3. Additives

As mentioned above, a stable SEI is essential for the performance of LIBs, especially in cold conditions. However, the deteriorated charge transfer kinetics on the SEI, including the dissolution process and the transport of Li^+^, seriously impact the low-temperature properties of LIBs [135]. Additives can change the composition and structure of the SEI during cycling, thus representing an important approach to overcome the barriers at low temperature. Additives mainly play two roles in the electrolyte: inhibiting the growth of the detrimental SEI and increasing the ionic conductivity. A small dose of additives can significantly enhance the properties of LIBs. To design a stable and protective interfacial layer which can adapt to low temperature, several additives have been explored. Among all, the FEC used in PC-based electrolytes and the VC used in EC-based electrolytes are the most common additives [136,137,138,139,140,141]. Considering their ideal film-forming ability and benefits to the solvation structure, fluoridated additives were added to the electrolyte system. In addition to the FEC, 2% fluorosulfonyl isocyanate (FI) was introduced into 1 M LiPF_6_ EC/DMC (1:1); the graphite/Li batteries with such fluoridated additives exhibited higher capacity at all rates compared to the pristine electrolyte. At −20 °C, the coin cells with FI could cycle with a current rate of 0.2 C, while the cells employed in the initial electrolyte lost all capacity after five cycles at 0.1 C [142]. As reported by Yang’s group, the addition of 1% 2,3,4,5,6-pentafluorophenyl methanesulfonate (PFPMS) increased the capacity retention from 55.0% to 66.3% at 0.5 C and reduced the polarization at −20 °C [143]. In addition to being employed as a solvent, DMS can be used as an additive. As shown in Figure 8a, the graphite with baseline electrolyte could not form a protective SEI, resulting in decomposition during cycling, while a uniform and stable SEI was constructed in the electrolyte containing DMS due to its weak combination with Li^+^. Such a modification of the SEI helped to reduce the impedance and suppress the decomposition of the electrolyte, further optimizing the performance at low temperature [144]. Various Li salts can also be employed as additives. Chen et al. [145] introduced 3% LiPO_2_F_2_ into the 1 M LiPF_6_ EC/EMC/DEC (3:5:2) electrolyte. The discharge capacity at −10 °C and 0.1 C of NCM/Li assembled with the optimized electrolyte increased from 110 to 133 mA·h·g^−1^. The battery provided high capacity retention of about 94% after 200 cycles at −10 °C, while there was nearly no capacity left in the cell without LiPO_2_F_2_ (Figure 8b). The improvement was attributed to the better film-forming performance and the stabler SEI, which could suppress the R_ct_ and promote the transport of Li^+^. Liao et al. [146] showed that lithium difluorobis(oxalato) phosphate (LiDFBOP) could be used as an additive; LiNi_0.5_Co_0.2_Mn_0.3_O_2_/graphite cells with LiDFBOP delivered 49% of their room-temperature capacity at −30 °C, while the batteries without additive only retained 14% capacity. Moreover, additives have also been employed to solve compatibility problems in LIB systems. PC has been selected for low-temperature batteries due to its low freezing point and strong solvation ability. However, when cycled with a graphite anode, the PC-based electrolyte may induce the undesired process of Li^+^/PC solvent co-intercalation, leading to a deleterious effect. Inorganic salt CsPF_6_ was confirmed to alleviate this dilemma when used as the additive. Li et al. added 0.05 M CsPF_6_ to 1 M LiPF_6_ EC/PC/EMC (1:1:8), and a high capacity retention close to 70% was delivered in the LiNi_0.8_Co_0.15_Al_0.05_O_2_/graphite battery at −40 °C, significantly greater than the 20% retention demonstrated by the pristine electrolyte [147]. Qian and coworkers [148] indicated that erythritol bis(carbonate) (EBC) with a two-EC-like structure could improve the performance of LIBs at −20 °C as an additive due to its weak solvation with Li^+^; it is expected to be further researched (Figure 8c).

Compared with electrode materials, the modification of the electrolyte is a more efficient and cost-effective strategy to tune the low-temperature performance. The results of notable studies on low-temperature electrolytes are outlined in Table 3. Considering the goals of high conductivity, low viscosity, weak solvation structure, and low freezing point, the design of a low-temperature electrolyte is considered as the key to enable efficient LIB operation. Aiming to find an ideal electrolyte, the strategy of using a cosolvent is one of the most successful ways to optimize the performance at low temperature, especially when coupled with other methods. Furthermore, fluorinated solvents and novel solvents should be given more attention as promising alternatives. Noteworthily, despite their apparent promise, most new solvents exhibit undesirable compatibility with anode. Studies on Li salt modifications are relatively rare compared to studies on solvent modifications. Great advancements have been achieved in the field of blended-salt electrolytes at room temperature, and such an approach has proven promising in low-temperature LIB applications. High-concentration electrolytes have introduced a promising framework to enhance the properties of LIBs, and the employment of diluents make it possible to apply such electrolytes at low temperature. Blended-salt and high-concentration electrolytes also present unique barriers to large-scale application such as the inevitable extra cost and complex interaction, which should be addressed in future efforts. In addition, since being proposed, additives have played a vital role in the field of low-temperature LIBs, demonstrating their effectiveness in optimizing performance. Thus, more low-temperature additives are expected to be exploited for the design of reliable electrolytes. In addition to endeavors in the battery chemical field, the design of low-temperature electrolytes requires joint effort in fields such as computation and molecular modification.

## 5. Summary and Prospects

Low-temperature degradation of the electrochemical properties of LIBs has become a major obstacle toward their widespread implementation. When implemented below 0 °C, LIBs suffer from problems such as reduced Li^+^ transport properties, dramatic electrode polarization, Li planting, and Li dendrites, incurring a loss of energy density, life deterioration, and potential safety issues. However, several areas including electric vehicles, as well as space, subsea, and military applications, have put forward particular requirements for high-energy LIBs in extreme environments with the rapid development and increased applicability of LIBs. Therefore, it is urgent to address the critical barriers and improve the low-temperature performance of LIBs. Over the past decades, extensive studies have been carried out on low-temperature LIBs, and several breakthroughs have been achieved. Some of the notable and attractive achievements in electrodes and electrolytes mentioned in this review are outlined in Figure 9. This review attempted to summarize the recent research progress and the main strategies for the optimization of low-temperature LIBs to stimulate more interest in the application of LIBs in extreme conditions.

At present, studies on low-temperature cathode materials have mainly focused on the methods of surface coating, particle modification, and external element doping. As the most common approach, a small amount of coating can substantially exploit the advantages of the high ionic conductivity and stability of the coating material, which can promote low-temperature performance without harming the pristine cathode. The particle size, purity, and morphology of the cathode exert a substantial influence on its internal performance, and these aspects are highly related to the conductivity, structure stability, specific surface area, and the transport path of Li^+^. By reducing the particle size, purifying the material, and modifying the morphology, the transport process and the (de)intercalation of lithium ions can be effectively improved. Doping with external elements to purposefully improve the performance at low temperature as a function of the synergistic effect between different elements is a promising strategy. As stated before, the majority of current endeavors related to low-temperature cathodes have focused on the optimization and modification of common materials such as LNMO, LiFePO_4_, and LiCoO_2_. Although some progress has been achieved, it has been limited by the complex experimental methods and inevitable side reactions. A great variety of innovative LIBs cathodes have been proposed in recent years, whereas studies on their properties at low temperature are still lacking. Moreover, high-voltage cathodes below 0 °C have not been well explored and should be further investigated. Future studies on low-temperature LIBs cathodes should continue to emphasize cathodes with a high rate capability and energy density.

At low temperature, on the anode side, graphite, as a representative material, suffers from a low Li^+^ diffusion coefficient and fatal Li dendrites. Coating, particle modification, element doping, and prelithiation are applied to overcome this deficiency. Despite some inspiring progress, which has enhanced the performance to a certain extent, graphite struggles to be employed on a large scale at low temperature due to the intractable issue of Li plating caused by the low voltage. Therefore, efforts have been made to find ideal low-temperature anode materials with great Li^+^ transport capacity, low polarization, and no plating effect. Ti-based materials are regarded as attractive anode candidates due to their satisfactory capability of Li^+^ (de)intercalation and small volume expansion. The high operating voltage of over 1.5 V vs. Li/Li^+^ effectively eliminates the risk of Li plating and Li dendrites at the cost of energy density loss. Simultaneously, the alloy anodes represented by Sn and Si have gained a great deal of attention over recent years as high-capacity and fast-charging anodes. The advantages of their high capacity and special conversion mechanism give rise to their great prospects to address the undesirable performance at low temperature. However, the dramatic volume effect severely degrades the cycling ability, while potentially destabilizing the SEI and leading to more side reactions. In summary, low-temperature anodes represent a relatively new field, with the early outcomes exhibiting promising prospects for future practicality. However, none of the novel low-temperature anodes show the ability to replace graphite in terms of comprehensive performance. Hence, accelerating the development and investigation of new anodes with superior electrode kinetics, high voltage, and large capacity, including titanium-based materials and alloy anodes, is of great significance for the application of LIBs at low temperature.

The electrolyte contributes significantly to the performance of LIBs. Designing an electrolyte with great Li^+^ transport properties, low viscosity, a wide electrochemical window, a low freezing point, and desirable capability in the formation of a stable, thin, and low-impedance SEI is the primary goal in the field of low-temperature electrolytes. Due to the intrinsic chemical limits of a single substance, it is impossible to accomplish the optimal functionalities using a single solvent. Consequently, more complicated electrolytes are needed to satisfy the above requirements at low temperature. The strategies of implementing low-freezing-point and low-viscosity cosolvents, solvent fluorination, blended-salt electrolytes, high-concentration electrolytes, and appropriate additives have attracted significant attention, being conducive to the optimization of low-temperature electrolytes. However, the theory of the SEI system at low temperature has not yet reached a consensus, and more attention should be paid to the interface between electrolyte and electrode, including the solvation structure of Li^+^ and the construction of the SEI, which seem to exert a more critical influence on LIBs at low temperature. Accordingly, future design of the ideal low-temperature electrolyte needs to consider both macro- and micro-perspectives.

Despite decades of successful achievements in both electrodes and electrolytes, considerable challenges remain that should be explored in depth in the future. Typically, high-performance LIBs are founded on the synergy of internal components. Consequently, it is necessary to pay special attention to the compatibility between each component. For instance, the matching of the cathode and anode, the compatibility of the electrode and electrolyte, and potential side reactions must be comprehensively considered in order to propose LIBs that can be widely employed in low-temperature environments with successful implementation. Moreover, the complete evaluation of both the thermal and the mechanical safety of low-temperature LIBs has not been realized, which may pose potential risks for future applications.

## Figures and Tables

**Figure 1 materials-15-08166-f001:**
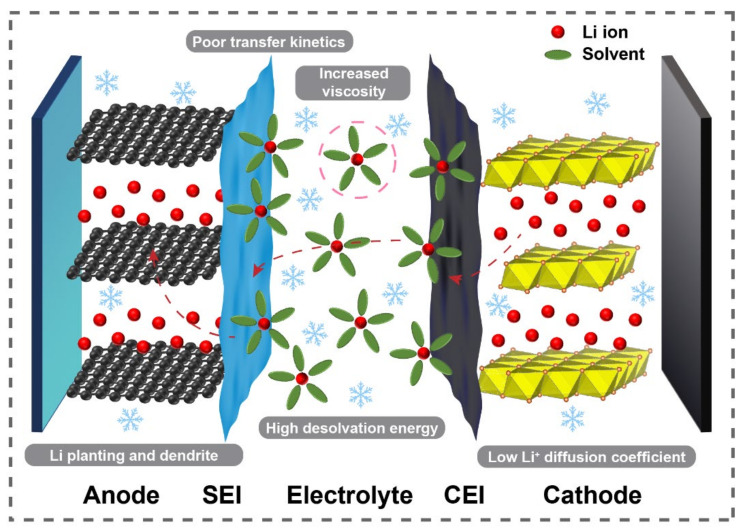
Schematic diagram of the problems in low-temperature LIBs.

**Figure 2 materials-15-08166-f002:**
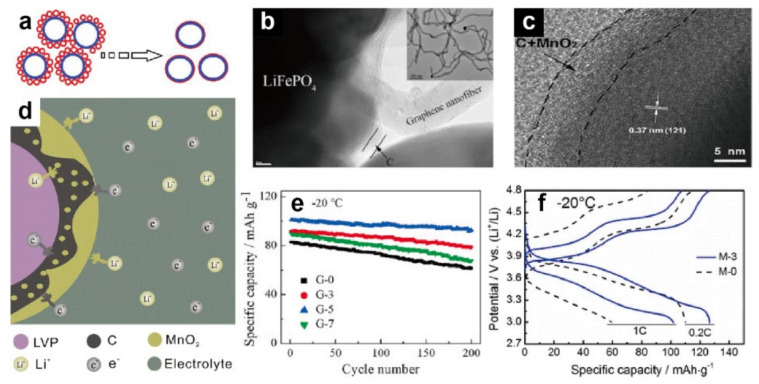
(**a**) Schematic diagram of carbon coating [44]. Reproduced with permission from Electrochimica Acta; published by Elsevier, 2011; (**b**) TEM image of LiFePO_4_/C with bridge network and graphene nanofiber as the insert [48]. Reproduced with permission from Electrochimica Acta; published by Elsevier, 2016; (**c**) HRTEM image of MnO_2_-coated LVP/C cathode, and (**d**) schematic illustration of its structure and diffusion path [50]. Reproduced with permission from Journal of Materials Science; published by Springer, 2017; (**e**) cycling performance of LiFePO_4_/C with different contents of coating at 1 C and −20 °C, where G−0, G−3, and G−7 represent the composites with 0 wt.%, 3 wt.%, and 7 wt.% graphene nanofibers, respectively [48]. Reproduced with permission from Electrochimica Acta; published by Elsevier, 2016; (**f**) first charge and discharge curves of M−3 and M−0 at −20 °C [50]. Reproduced with permission from Journal of Materials Science; published by Springer, 2017.

**Figure 3 materials-15-08166-f003:**
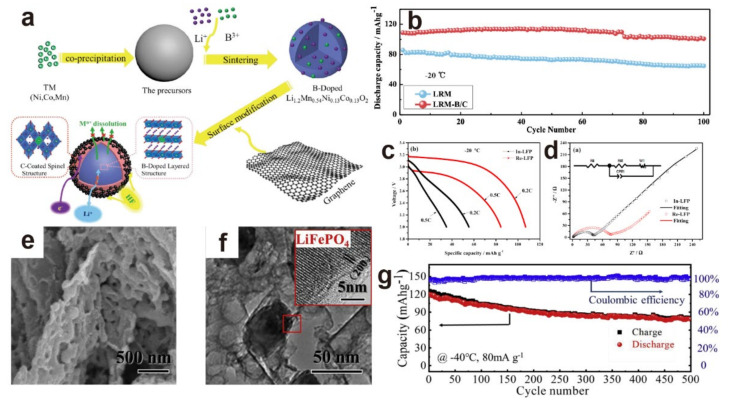
(**a**) Synthesis method and structure of the B-doped and carbon-modified Li_1.2_Mn_0.54_Ni_0.13_Co_0.13_O_2_; (**b**) corresponding cycling performance at 1 C under −20 °C [60]. Reproduced with permission from ACS Sustainable Chemistry & Engineering; published by American Chemical Society, 2020; (**c**) the first discharge curves of In-LFP and Re-LFP at −20 °C; (**d**) the EIS pattern of In-LFP and Re-LFP [67]. Reproduced with permission from Journal of Alloys and Compounds; published by Elsevier, 2016; (**e**,**f**) SEM image and TEM image of the coral-shaped LiFePO_4_/graphene; (**g**) cycling performance of half-cell at −40 °C with 80 mA·g^−1^ [69]. Reproduced with permission from Energy Storage Materials; published by Elsevier, 2019.

**Figure 4 materials-15-08166-f004:**
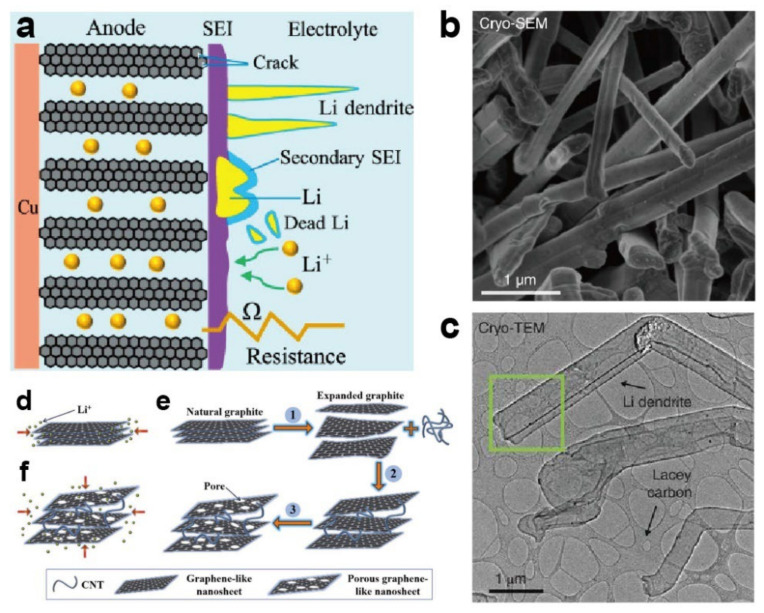
(**a**) Problems with graphite anode at low temperature [73]. Reproduced with permission from Journal of Power Sources; published by Elsevier, 2020; (**b**,**c**) Cryo-SEM image and Cryo-TEM image of Li dendrites [78]. Reproduced with permission from Science; published by American Association for the Advancement of Science, 2017; schematic diagram of Li ion intercalation process in (**d**) conventional graphite and (**e**) PGN; (**f**) synthesis of PGN/CNT material [79]. Reproduced with permission from Journal of Power Sources; published by Elsevier, 2019.

**Figure 5 materials-15-08166-f005:**
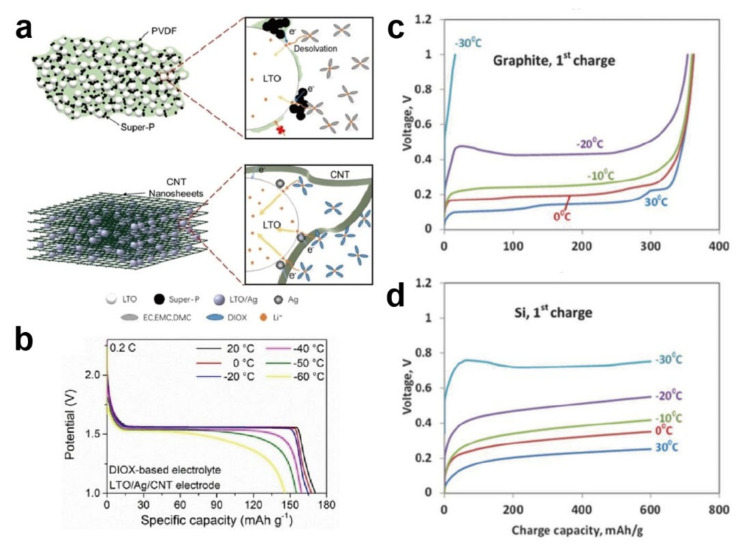
(**a**) Schematic illustration of common LTO with conductive agent (Super−P), polymer binder (PVDF), and schematic illustration of binder-free LTO/Ag/CNT, showing their enlarged local parts during lithiation on the right; (**b**) charge voltage profiles under different temperatures at 0.2 C of LTO/Ag/CNT electrode [81]. Reproduced with permission from ChemElectroChem; published by Wiley-VCH Verlag GmbH & Co. KGaA, 2020; (**c**,**d**) the first charge curves at 0.25 C and different temperatures of silicon anode and graphite anode [82]. Reproduced with permission from Journal of the Electrochemical Society; published by IOP Publishing Ltd., 2016.

**Figure 6 materials-15-08166-f006:**
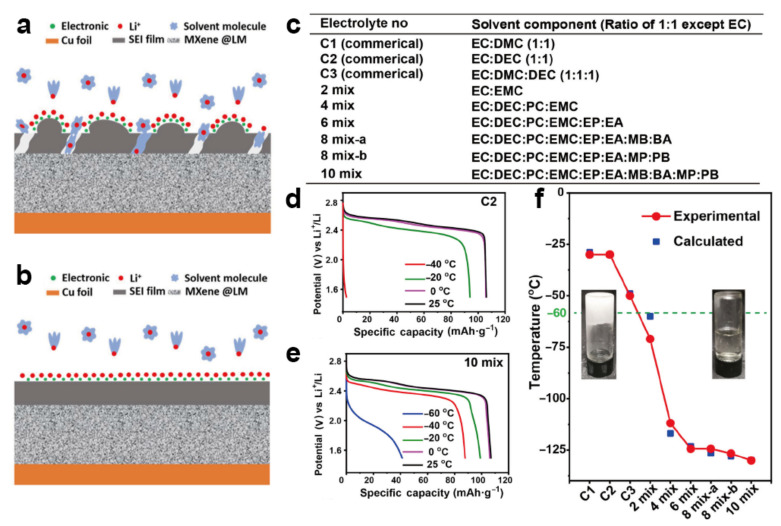
Schematic diagram of the electrode surface film before (**a**) and after (**b**) using the MB electrolyte at low temperature [109]. Reproduced with permission from Chinese Journal of Chemistry; published by Wiley-VCH Verlag GmbH & Co. KGaA, 2021; (**c**) solvent compositions of commercial electrolytes and the designed multicomponent high-entropy electrolytes with 1 M LiPF_6_ and fluoroethylene carbonate (FEC) as additives; (**d**,**e**) discharge curves of LiMn_2_O_4_/LTO full cell with C2 and 10 electrolyte mixtures at 0.1 C under different temperatures; (**f**) the freezing point of various electrolytes [111]. Reproduced with permission from CCS Chemistry; published by Chinese Chemical Society, 2021.

**Figure 7 materials-15-08166-f007:**
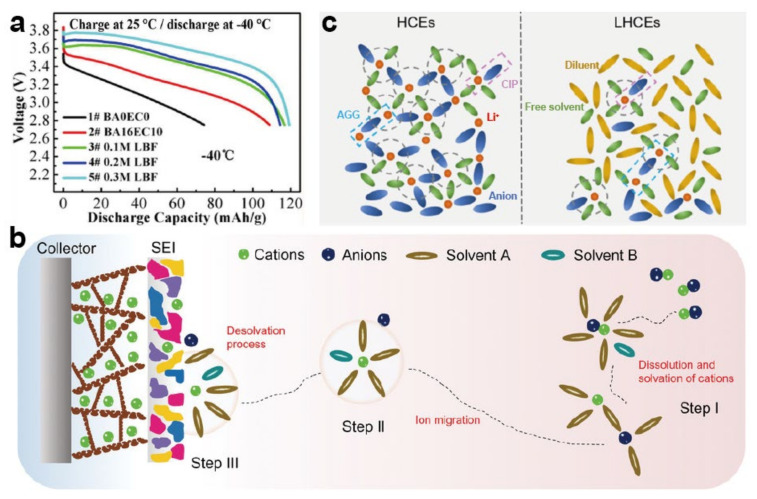
(**a**) Charge and discharge curves of NCM811 cells with different electrolytes at −40 °C [16]. Reproduced with permission from Chemical engineering journal; published by Elsevier, 2021; (**b**) illustration of the solvation process during operation [127]. Reproduced with permission from Chinese Journal of Chemistry; published by Wiley, 2022; (**c**) schematic of the solvation structures of HCEs and LHCEs [128]. Reproduced with permission from ACS Applied Materials & Interfaces; published by American Chemical Society, 2022.

**Figure 8 materials-15-08166-f008:**
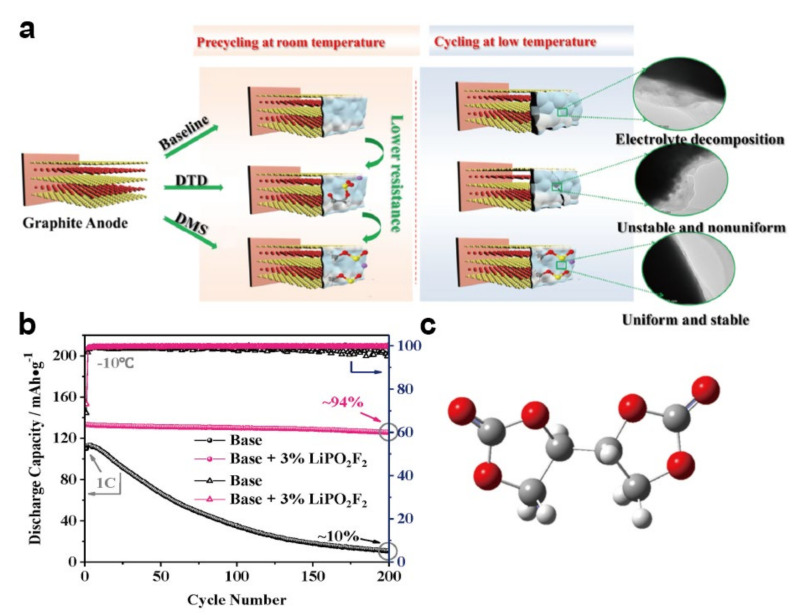
(**a**) Schematic diagram of the SEI on a graphite anode in an electrolyte with different additives [144]. Reproduced with permission from ACS Applied Materials & Interfaces; published by American Chemical Society, 2019; (**b**) cycling performance of NCM/Li in electrolyte with and without LiPO_2_F_2_ [145]. Reproduced with permission from Electrochimica Acta; published by Elsevier Ltd., 2018; (**c**) molecular structure of EBC [148]. Reproduced with permission from Energy Storage Materials; published by Elsevier, 2021.

**Figure 9 materials-15-08166-f009:**
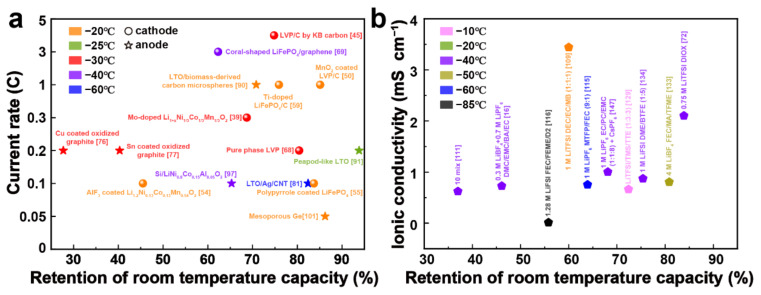
Notable research progresses in low-temperature LIBs: (**a**) electrodes; (**b**) electrolytes.

**Table 1 materials-15-08166-t001:** Summary of the low-temperature electrochemical properties of different notable cathodes.

Material	Cell Type	Capacity	Stability	Theoretical Capacity	Mass Loading	Ref.
LVP/C from KB carbon	Half-cell	92 mA·h·g^−1^ (−30 °C, 4 C)	99% after 500 cycles (−30 °C, 4 C)	132 mA·h·g^−1^	1–2 mg·cm^−2^	[45]
LiFePO_4_/C	Half-cell	109.2 mA·h·g^−1^ (−20 °C, 0.1 C)	91% after 50 cycles (0 °C, 0.1 C)	170 mA·h·g^−1^	N/A	[46]
LiFePO_4_/CNT with conductive network	Half-cell	90 mA·h·g^−1^ (−10 °C, 1 C)	N/A	170 mA·h·g^−1^	3.7–4.2 mg·cm^−2^	[47]
LiFePO_4_/graphene nanofibers	Half-cell	124.4 mA·h·g^−1^ (−20 °C, 0.1 C)	92% after 200 cycles (−20 °C, 1 C)	170 mA·h·g^−1^	N/A	[48]
LiFePO_4_/C from fructose and calcium lignosulfonate	Half-cell	109.4 mA·h·g^−1^ (−20 °C, 0.5 C)	N/A	170 mA·h·g^−1^	N/A	[49]
Li_2_O·B_2_O_3_-coated NCM	Half-cell	101.9 mA·h·g^−1^ (−40 °C, 0.2 C)	70.6% after 50 cycles (−20 °C, 0.2 C)	280 mA·h·g^−1^	8.2 mg·cm^−2^	[18]
MnO_2_-coated LVP/C	Half-cell	102 mA·h·g^−1^ (−20 °C, 1 C)	N/A	197 mA·h·g^−1^	2.5 mg·cm^−2^	[50]
CeO_2_-coated LVP/C	Half-cell	103.3 mA·h·g^−1^ (−20 °C, 1 C)	95.6% after 50 cycles (−20°C, 5 C)	197 mA·h·g^−1^	N/A	[51]
CeO_2_-coated LiFePO_4_/C	Half-cell	99.7 mA·h·g^−1^ (−20 °C, 0.1 C)	N/A	170 mA·h·g^−1^	N/A	[52]
AlF_3_-coated Li_1.2_Ni_0.13_Co_0.13_Mn_0.54_O_2_	Half-cell	109.3 mA·h·g^−1^ (−20 °C, 0.1 C)	N/A	N/A	N/A	[54]
Polypyrrole-coated LiFePO_4_	Half-cell	128 mA·h·g^−1^ (−20 °C, 0.1 C)	N/A	170 mA·h·g^−1^	N/A	[55]
Mo-doped Li_1+x_Ni_1/3_Co_1/3_Mn_1/3_O_2_	Half-cell	65.4 mA·h·g^−1^ (−30 °C, 5 C)	N/A	280 mA·h·g^−1^	N/A	[39]
Ti-doped LiFePO_4_/C	Full cell	112.8 mA·h·g^−1^ (−20 °C, 1 C)	N/A	170 mA·h·g^−1^	N/A	[59]
B-doped Li_1.2_Mn_0.54_Ni_0.13_Co_0.13_O_2_	Half-cell	101.9 mA·h·g^−1^ (−20 °C, 0.2 C)	93.4% after 50 cycles (− 20 °C, 0.2 C)	N/A	N/A	[60]
P-doped LiFePO_4_/C -	Half-cell	82.7 mA·h·g^−1^ (−40 °C, 0.1 C)	N/A	170 mA·h·g^−1^	1.092 mg·cm^−2^	[61]
Co-doped Li_1.2_Ni_0.2_Mn_0.6_O_2_	Half-cell	111.3 mA·h·g^−1^ (−20 °C, 0.1 C)	58.4% after 40 cycles (−20 °C, 0.1 C)	N/A	N/A	[62]
Re-LFP	Half-cell	107 mA·h·g^−1^ (−20 °C, 0.2 C)	N/A	170 mA·h·g^−1^	N/A	[67]
Pure phase LVP	Half-cell	90 mA·h·g^−1^ (−30 °C, 0.2 C)	80% after 1000 cycles (−10 °C, 10 C)	132 mA·h·g^−1^	2 mg·cm^−2^	[68]
Coral-shaped LiFePO_4_/graphene	Full cell	101 mA·h·g^−1^ (−40 °C, 3 C)	91% after 20 cycles (−40 °C, 3 C)	170 mA·h·g^−1^	8–10 mg	[69]

**Table 2 materials-15-08166-t002:** Summary of the low-temperature electrochemical properties of different notable anodes.

Material	Cell Type	Capacity	Stability	Theoretical Capacity	Mass Loading	Ref.
BNG	Half-cell	135.8 mA·h·g^−1^ (−20 °C, 0.1 C)	N/A	N/A	N/A	[75]
Cu-coated oxidized graphite	Half-cell	103 mA·h·g^−1^ (−30 °C, 0.2 C)	N/A	372 mA·h·g^−1^	1.5–2 mg·cm^−2^	[76]
Sn-coated oxidized graphite	Half-cell	152 mA·h·g^−1^ (−30 °C, 0.2 C)	N/A	372 mA·h·g^−1^	2–3 mg·cm^−2^	[77]
Multilayer crystalline graphene	Half-cell	130 mA·h·g^−1^ (−30 °C, 0.05 A g^−1^)	N/A	N/A	0.75 mg·cm^−2^	[80]
TiO_2_/TiN/graphene	Half-cell	211 mA·h·g^−1^ (−20 °C, 0.1 A g^−1^)	93% after 500 cycles (−20 °C, 1 A·g^−1^)	N/A	1–1.5 mg·cm^−2^	[6]
PGN/CNT	Half-cell	180 mA·h·g^−1^ (−40 °C, 0.1 C)	N/A	N/A	2 mg·cm^−2^	[79]
LTO/Ag/CNT	Half-cell	140 mA·h·g^−1^ (−60 °C, 0.2 C)	N/A	175 mA·h·g^−1^	4 mg·cm^−2^ (electrode)	[81]
Fe/Fe_3_C/CNFs	Half-cell	380 mA·h·g^−1^ (−15 °C, 0.2 A g^−1^)	N/A	N/A	1.5 mg·cm^−2^	[83]
Smaller primary LTO	Full cell	109 mA·h·g^−1^ (−20 °C, 1 C)	N/A	175 mA·h·g^−1^	5.9−6.4 mg·cm^−2^	[89]
LTO/biomass-derived carbon microspheres	Half-cell	150 mA·h·g^−1^ (−20 °C, 1 C)	N/A	N/A	N/A	[90]
Peapod-like LTO	Half-cell	167 mA·h·g^−1^ (−25 °C, 0.2 C)	96% after 500 cycles (−25 °C, 10 C)	175 mA·h·g^−1^	12 mg·cm^−2^	[91]
Si	Half-cell	600 mA·h·g^−1^ (−30 °C, 0.25 C)	N/A	3580 mA·h·g^−1^	1.6 mg·cm^−2^	[82]
Si/LiNi_0.8_Co_0.15_Al_0.05_O_2_	Pouch cell	707 mA h (−40 °C, 0.1 C)	100% after 40 cycles (−40 °C, 0.1 C)	N/A	N/A	[97]
Mesoporous Ge	Half-cell	1178 mA·h·g^−1^ (−20 °C, 0.5 C)	50% after 50 cycles (−20 °C, 0.5 C)	1624 mA·h·g^−1^	N/A	[101]

**Table 3 materials-15-08166-t003:** Summary of the low-temperature electrochemical properties of recently studied electrolytes.

Electrolyte	Cell	Capacity	Stability	Theoretical Capacity	Mass Loading	Ref.
1 M LiTFSI DEC/EC/MB (1:1:1)	Li/MXene@LM	390 mA·h·g^−1^ (−20 °C, 200 mA·g^−1^)	N/A	N/A	N/A	[109]
1 M LiPF_6_ EC/EMC/PC/DMC (1.8:3:0.3:3.5) + VC + LiBOB	NCM/graphite	0.74 mA·h (−20 °C, 5 C)	N/A	280 mA·h·g^−1^	6.8 mg·cm^−2^	[110]
10 mixtures	LiMn_2_O_4_/LTO	88 mA·h·g^−1^ (−40 °C, 0.1 C)	about 90% after 40 cycles (−40 °C, 0.2 C)	148 mA·h·g^−1^	N/A	[111]
1 M LiPF_6_ MP/VC (95:5)	240 mA h NCM/graphite	590 mW·h (−14 °C, 0.1 C)	N/A	N/A	N/A	[112]
1 M LiPF_6_ MTFP/FEC (9:1)	NCM811/Li	133 mA·h·g^−1^ (−60 °C, 0.1 C)	N/A	200 mA·h·g^−1^	1.3 mA·h·cm^−2^	[115]
1.28 M LiFSI FEC/FEME/D2	LiNi_0.8_Co_0.15_Al_0.05_O_2_/Li	96 mA·h·g^−1^ (−85 °C, 0.1 C)	over 90% after 450 cycles (−20 °C, 1/3 C)	N/A	N/A	[116]
1 M LiDFOB FEC/IZ (1:10)	Li/graphite	187.5 mA·h·g^−1^ (−20 °C, 0.1 C)	N/A	372 mA·h·g^−1^	3–4 mg·cm^−2^	[117]
0.9 M LiODFB SL/DMS (1:1)	LiFePO_4_/Li	80 mA·h·g^−1^ (−20 °C, 0.5 C)	95.57% after 50 cycles (−20 °C, 0.5 C)	170 mA·h·g^−1^	N/A	[118]
0.75 M LiTFSI DIOX	LTO/Li	Over 130 mA·h·g^−1^ (−40 °C, 0.1 C)	N/A	175 mA·h·g^−1^	8 mg·cm^−2^	[72]
0.3 M LiBF_4_ + 0.7 M LiPF_6_ DMC/EMC/BA/EC	NCM811/Li	About 86 mA·h·g^−1^ (−40 °C, 0.2 C)	N/A	280 mA·h·g^−1^	5.56 mg·cm^−2^	[16]
LiTFSI/TMS/TTE (1:3:3)	NCM/Li	over 100 mA·h·g^−1^ (−10 °C, 0.2 C)	N/A	280 mA·h·g^−1^	1.5 mA·h·cm^−2^	[129]
4 M LiBF_4_ FEC/MA/TFME	LNMO/Li	over 100 mA·h·g^−1^ (−50 °C, 0.2 C)	93.8% after 100 cycles (−40 °C, 0.1 C)	147 mA·h·g^−1^	2.58 mg·cm^−2^	[133]
1 M LiFSI DME/BTFE (1:5)	NCM811/Li	153 mA·h·g^−1^ (−40 °C, 0.1 C)	100% after 200 cycles (−40 °C, 0.1 C)	280 mA·h·g^−1^	N/A	[134]
1 M LiPF_6_ EC/DMC (1:1) + FI	Li/graphite	30 mA·h·g^−1^ (−20 °C, 0.1 C)	N/A	372 mA·h·g^−1^	3.31 mg·cm^−2^	[142]
1 M LiPF_6_ EC/EMC (1:2) + PFPMS	LiNi_0.5_Mn_0.2_Co_0.3_O_2_/graphite	over 1350 mA·h (−20 °C, 0.2 C)	N/A	N/A	N/A	[143]
1 M LiPF_6_ EC/EMC/DEC (3:5:2) + LiPO_2_F_2_	NCM/Li	133 mA·h·g^−1^ (−10 °C, 0.1 C)	94% after 200 cycles (−10 °C, 0.1 C)	280 mA·h·g^−1^	3.9 ± 0.1 mg·cm^−2^	[145]
1 M LiPF_6_ EC/EMC (1:2) + LiDFBOP	LiNi_0.5_Co_0.2_Mn_0.3_O_2_/graphite	N/A	93% after 200 cycles (−20 °C, 0.5 C)	N/A	31.18 mg·cm^−2^ (electrode)	[146]
1 M LiPF_6_ EC/PC/EMC (1:1:8) + CsPF_6_	LiNi_0.8_Co_0.15_Al_0.05_O_2_/graphite	112.3 mA·h·g^−1^ (−40 °C, 0.1 C)	N/A	275 mA·h·g^−1^	1.5 mA·h·cm^−2^	[147]

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
