# Peer review of "Lithium-Ion Batteries under Low-Temperature Environment: Challenges and Prospects"

_materials, 2022, doi:10.3390/ma15228166_

Round 1

Reviewer 1 Report

The authors have worked on a review for Lithium-Ion Batteries under Low-Temperature Environment: Challenges and Prospects. There have been numerous review articles on similar areas thus, the manuscript requires modification before consideration for publication.

1) Abstract: Kindly elaborate more on the challenges and prospects in the abstract section.

Introduction: The section is well written; however, kindly discuss the challenges of LiB at different temperatures, i.e., Room temperature, low and high temperature, and their stability issue.

There has been enormous work on carbon material for Lib. Thus, the section "3.1. Carbon-based anodes" should be individually sub-sectioned into different carbon base anodes like Graphene based anodes or Carbon nanotube-based anodes.

Silicon has been a great anodic material due to its large capacity but also sustained issues with volume shrinkage and etc.. Thus, a section based on Silicon as an anode must be added to the manuscript.

1) Favors, Zachary, et al. "Scalable synthesis of nano-silicon from beach sand for long cycle life Li-ion batteries." Scientific reports 4.1 (2014): 1-7.

3.2. Ti-based anodes: Kindly add the article's recent literature on Titania-based anodes. Here are a couple of suggestions:

1) Khanna, Sakshum, et al. "In-situ preparation of titania/graphene nanocomposite via a facile sol–gel strategy: A promising anodic material for Li-ion batteries." Materials Letters 300 (2021): 130143.

2) Sopha, Hanna, et al. "TiO2 nanotube layers decorated by titania nanoparticles as anodes for Li-ion microbatteries." Materials Chemistry and Physics 276 (2022): 125337.

The result for various literature needs to be added with a graph, comparing the result under similar conditions to have more clarity for readers. 

An individual table of literature for cathode, anode, and electrolyte should be added for better understanding. 

There is unclarity for some sentences due to grammatical errors. Kindly improve the grammar in the revised version.

The manuscript needs to add more information, results, and comparison results for readers to understand the challenges and prospects in LiB.

Reviewer 2 Report

This manuscript deals with the application fields of Li-ion batteries and the problems that arise when they are used at low temperatures. To reduce these problems, numerous possibilities are presented starting with the influence of coatings on the cathode properties or the choice of the alloy composition of the anode up to the solvents used and their properties. The authors have also highlighted the influence of the various modifications on storage capacity and cycle stability at low temperatures.

Nevertheless, a few things need to be improved for the paper to be accepted.

·      The quality of the figures is low, e.g. Fig. 3 is partly too small and hard to read

·       Some of the graphs should be better explained, e.g. in figure 1, all symbols should be labeled.

·       The word choice and the sentence structures should be revised again and single terms should be defined more precisely so that it fits into the context. e.g. p. 1 line 25…“space”…it is not clear.

·       Some terms should be described more precisely or formulated differently, for example: as mentioned in the abstract, LiBs are “environmental friendliness” but, among other things, the mining of Li is not environmentally friendly. Therefore it is better to define it carefully.

·       Recommendation: It would be nice to see if the work can provide some assessment of the technical applicability and the costs associated with the manufacture/production. Is this technically feasible on an industrial scale?

Reviewer 3 Report

The review covers the main current trends in the study of lithium-ion batteries, the properties of electrode materials and electrolyte on stability at low operating temperatures. In general, the work is full-fledged and can be presented in the journal Materials, taking into account the correction of arising issues.

1.Since quite a lot of cross-cutting abbreviations and acronyms are used, it is recommended to add a section with a list of abbreviations and their explanation.

2. Page 5, line 191 "By reducing the size of material, the discharge capacity has increased by about 44%. "It is worth clarifying the sentence. It is not the size of the material that is reduced, but the particle/grain size of the material.

3. It is necessary to specify the text from the work [67], p.5 lines 193-196.  The impedance of the In-LFP sample is lower than that of the pure sample; this can be seen from the impedance spectra, Figure 3f. The authors of [67] attribute this to the presence of highly conductive impurities

4. Page 7, line286 it should be clarified that the capacitance loss of the Ti-based anode is significant, dropping by 40% of room temperature at low -80C.

5. On page 11 you should add the abbreviation EC solvent. On page 13, figure 6c, decipher the abbreviation BA solvent. Decipher the abbreviation FEC in the caption of figure 6d-e. Remove the title of the publication [101] from the caption of Figure 6f.

6. The CEI abbreviation from figure 1 should be deciphered in the text.

7.Figure 2a should be mentioned in the text and G0-G7 in Figure 2e should be transcribed.

8. For figure 2f, the notation M-3 and M-0 should be explained and a reference to the original source should be provided

Reviewer 4 Report

Comments to the authors

The manuscript entitled “Lithium-Ion Batteries under Low-Temperature Environment: Challenges and Prospects” has been submitted by the author. However, there are still major concerns that need to be addressed before publication. Therefore, I recommend this work could be published after the major revision.

  1. The same kind of reviews (https://doi.org/10.1016/j.etran.2021.100145, https://onlinelibrary.wiley.com/doi/full/10.1002/aesr.202100039... etc) has already reported in the literature. So, author should rewrite the introduction part with stat of art.

2.      What happened to Li-ion battery in low temperature? Explain in detail

3.      Author should mention the performance of Li-ion battery in cold condition with electrode weight.

  1. It is recommended to mention the values of theoretical capacities of the reported electrodes.
  2. Give the mechanism, how does temperature affect the Li-ion battery.
  3. Revision and modifications are required before acceptance of this work.
  4. English should be checked through the whole manuscript because there are many spelling and grammatical errors in the paper.

Round 2

Reviewer 1 Report

The authors have made significant modification and provided proper response in the manuscript. Thus the manuscript can be acceptable in present form.

Reviewer 4 Report

Authors has modified the paper according to the comments. Now review paper is well within the shape and should be accepted.